# Improved and Generalized Upper Bounds on the Complexity of Policy Iteration

**Bruno Scherrer**

Inria, Villers-lès-Nancy, F-54600, France
Université de Lorraine, LORIA, UMR 7503, Vandoeuvre-lès-Nancy, F-54506, France
`bruno.scherrer@inria.fr`

## Abstract

Given a Markov Decision Process (MDP) with $n$ states and $m$ actions per state, we study the number of iterations needed by Policy Iteration (PI) algorithms to converge to the optimal $\gamma$-discounted optimal policy. We consider two variations of PI: Howard's PI that changes the actions in all states with a positive advantage, and Simplex-PI that only changes the action in the state with maximal advantage. We show that Howard's PI terminates after at most $n(m-1)\left\lceil \frac{1}{1-\gamma}\log\left(\frac{1}{1-\gamma}\right)\right\rceil = O\left(\frac{nm}{1-\gamma}\log\left(\frac{1}{1-\gamma}\right)\right)$ iterations, improving by a factor $O(\log n)$ a result by [3], while Simplex-PI terminates after at most $n^2(m-1)\left(1+\frac{2}{1-\gamma}\log\left(\frac{1}{1-\gamma}\right)\right) = O\left(\frac{n^2 m}{1-\gamma}\log\left(\frac{1}{1-\gamma}\right)\right)$ iterations, improving by a factor $O(\log n)$ a result by [11]. Under some structural assumptions of the MDP, we then consider bounds that are independent of the discount factor $\gamma$: given a measure of the maximal transient time $\tau_t$ and the maximal time $\tau_r$ to revisit states in recurrent classes under all policies, we show that Simplex-PI terminates after at most $n^2(m-1)\left(\lceil \tau_r \log(n\tau_r)\rceil + \lceil \tau_r \log(n\tau_t)\rceil\right)\left[(m-1)\lceil n\tau_t \log(n\tau_t)\rceil + \lceil n\tau_t \log(n^2\tau_t)\rceil\right] = \tilde{O}\left(n^3 m^2 \tau_t \tau_r\right)$ iterations. This generalizes a recent result for deterministic MDPs by [8], in which $\tau_t \leq n$ and $\tau_r \leq n$. We explain why similar results seem hard to derive for Howard's PI. Finally, under the additional (restrictive) assumption that the state space is partitioned in two sets, respectively states that are transient and recurrent for all policies, we show that Howard's PI terminates after at most $n(m-1)\left(\lceil \tau_t \log n\tau_t\rceil + \lceil \tau_r \log n\tau_r\rceil\right) = \tilde{O}(nm(\tau_t + \tau_r))$ iterations while Simplex-PI terminates after $n(m-1)\left(\lceil n\tau_t \log n\tau_t\rceil + \lceil \tau_r \log n\tau_r\rceil\right) = \tilde{O}(n^2 m(\tau_t + \tau_r))$ iterations.

## 1 Introduction

We consider a discrete-time dynamic system whose state transition depends on a control. We assume that there is a **state space** $X$ of finite size $n$. At state $i \in \{1, .., n\}$, the control is chosen from a **control space** $A$ of finite size[1] $m$. The control $a \in A$ specifies the **transition probability** $p_{ij}(a) = \mathbb{P}(i_{t+1} = j | i_t = i, a_t = a)$ to the next state $j$. At each transition, the system is given a reward $r(i, a, j)$ where $r$ is the instantaneous **reward function**. In this context, we look for a stationary deterministic policy (a function $\pi : X \to A$ that maps

states into controls[2]) that maximizes the expected discounted sum of rewards from any state $i$, called the **value of policy** $\pi$ at state $i$:

$$v_\pi(i) := \mathbb{E}\left[\sum_{k=0}^{\infty} \gamma^k r(i_k, a_k, i_{k+1}) \,\middle|\, i_0 = i,\ \forall k \geq 0,\ a_k = \pi(i_k),\ i_{k+1} \sim \mathbb{P}(\cdot|i_k, a_k)\right]$$

where $\gamma \in (0,1)$ is a discount factor. The tuple $\langle X, A, p, r, \gamma \rangle$ is called a **Markov Decision Process (MDP)** [9, 1], and the associated problem is known as **optimal control.**

The **optimal value** starting from state $i$ is defined as

$$v_*(i) := \max_\pi v_\pi(i).$$

For any policy $\pi$, we write $P_\pi$ for the $n \times n$ stochastic matrix whose elements are $p_{ij}(\pi(i))$ and $r_\pi$ the vector whose components are $\sum_j p_{ij}(\pi(i)) r(i, \pi(i), j)$. The value functions $v_\pi$ and $v_*$ can be seen as vectors on $X$. It is well known that $v_\pi$ is the solution of the following Bellman equation:

$$v_\pi = r_\pi + \gamma P_\pi v_\pi,$$

that is $v_\pi$ is a fixed point of the affine operator $T_\pi : v \mapsto r_\pi + \gamma P_\pi v$. It is also well known that $v_*$ satisfies the following Bellman equation:

$$v_* = \max_\pi (r_\pi + \gamma P_\pi v_*) = \max_\pi T_\pi v_*$$

where the max operator is componentwise. In other words, $v_*$ is a fixed point of the nonlinear operator $T : v \mapsto \max_\pi T_\pi v$. For any value vector $v$, we say that a policy $\pi$ is **greedy with respect to the value** $v$ if it satisfies:

$$\pi \in \arg\max_{\pi'} T_{\pi'} v$$

or equivalently $T_\pi v = Tv$. With some slight abuse of notation, we write $\mathcal{G}(v)$ for any policy that is greedy with respect to $v$. The notions of optimal value function and greedy policies are fundamental to optimal control because of the following property: any policy $\pi_*$ that is greedy with respect to the optimal value $v_*$ is an **optimal policy** and its value $v_{\pi_*}$ is equal to $v_*$.

Let $\pi$ be some policy. We call **advantage with respect to** $\pi$ the following quantity:

$$a_\pi = \max_{\pi'} T_{\pi'} v_\pi - v_\pi = Tv_\pi - v_\pi.$$

We call the **set of switchable states of** $\pi$ the following set

$$S_\pi = \{i,\ a_\pi(i) > 0\}.$$

Assume now that $\pi$ is non-optimal (this implies that $S_\pi$ is a non-empty set). For any non-empty subset $Y$ of $S_\pi$, we denote $\text{switch}(\pi, Y)$ a policy satisfying:

$$\forall i,\ \text{switch}(\pi, Y)(i) = \begin{cases} \mathcal{G}(v_\pi)(i) & \text{if } i \in Y \\ \pi(i) & \text{if } i \notin Y. \end{cases}$$

The following result is well known (see for instance [9]).

**Lemma 1.** *Let $\pi$ be some non-optimal policy. If $\pi' = switch(\pi, Y)$ for some non-empty subset $Y$ of $S_\pi$, then $v_{\pi'} \geq v_\pi$ and there exists at least one state $i$ such that $v_{\pi'}(i) > v_\pi(i)$.*

This lemma is the foundation of the well-known iterative procedure, called Policy Iteration (PI), that generates a sequence of policies $(\pi_k)$ as follows.

$$\pi_{k+1} \leftarrow \text{switch}(\pi_k, Y_k) \text{ for some set } Y_k \text{ such that } \emptyset \subsetneq Y_k \subseteq S_{\pi_k}.$$

The choice for the subsets $Y_k$ leads to different variations of PI. In this paper we will focus on two specific variations:

- When for all iterations $k$, $Y_k = S_{\pi_k}$, that is one switches the actions in all states with positive advantage with respect to $\pi_k$, the above algorithm is known as Howard's PI; it can be seen then that $\pi_{k+1} \in \mathcal{G}(v_{\pi_k})$.
- When for all $k$, $Y_k$ is a singleton containing a state $i_k \in \arg\max_i a_{\pi_k}(i)$, that is if we only switch one action in the state with maximal advantage with respect to $\pi_k$, we will call it Simplex-PI[3].

Since it generates a sequence of policies with increasing values, any variation of PI converges to the optimal policy in a number of iterations that is smaller than the total number of policies $m^n$. In practice, PI converges in very few iterations. On random MDP instances, convergence often occurs in time sub-linear in $n$. The aim of this paper is to discuss existing and provide new upper bounds on the number of iterations required by Howard's PI and Simplex-PI that are much sharper than $m^n$.

In the next sections, we describe some known results—see [11] for a recent and comprehensive review—about the number of iterations required by Howard's PI and Simplex-PI, along with some of our original improvements and extensions.[4]

## 2 Bounds with respect to a Fixed Discount Factor $\gamma < 1$

A key observation for both algorithms, that will be central to the results we are about to discuss, is that the sequence they generate satisfies some contraction property[5]. For any vector $u \in \mathbb{R}^n$, let $\|u\|_\infty = max_{1 \leq i \leq n} |u(i)|$ be the max-norm of $u$. Let $\mathbb{1}$ be the vector of which all components are equal to 1.

**Lemma 2** (Proof in Section A). *The sequence $(\|v_* - v_{\pi_k}\|_\infty)_{k \geq 0}$ built by Howard's PI is contracting with coefficient $\gamma$.*

**Lemma 3** (Proof in Section B). *The sequence $(\mathbb{1}^T(v_* - v_{\pi_k}))_{k \geq 0}$ built by Simplex-PI is contracting with coefficient $1 - \frac{1-\gamma}{n}$.*

Though this observation is widely known for Howard's PI, it was to our knowledge never mentionned explicitly in the literature for Simplex-PI. These contraction properties have the following immediate consequence[6].

**Corollary 1.** *Let $V_{\max} = \frac{\max_\pi \|r_\pi\|_\infty}{1-\gamma}$ be an upper bound on $\|v_\pi\|_\infty$ for all policies $\pi$. In order to get an $\epsilon$-optimal policy, that is a policy $\pi_k$ satisfying $\|v_* - v_{\pi_k}\|_\infty \leq \epsilon$, Howard's PI requires at most $\left\lceil \frac{\log \frac{V_{\max}}{\epsilon}}{1-\gamma} \right\rceil$ iterations, while Simplex-PI requires at most $\left\lceil \frac{n \log \frac{n V_{\max}}{\epsilon}}{1-\gamma} \right\rceil$ iterations.*

These bounds depend on the precision term $\epsilon$, which means that Howard's PI and Simplex-PI are *weakly polynomial* for a fixed discount factor $\gamma$. An important breakthrough was recently achieved by [11] who proved that one can remove the dependency with respect to $\epsilon$, and thus show that Howard's PI and Simplex-PI are *strongly polynomial* for a fixed discount factor $\gamma$.

**Theorem 1** ([11]). *Simplex-PI and Howard's PI both terminate after at most $n(m - 1) \left\lceil \frac{n}{1-\gamma} \log \left( \frac{n^2}{1-\gamma} \right) \right\rceil$ iterations.*

The proof is based on the fact that PI corresponds to the simplex algorithm in a linear programming formulation of the MDP problem. Using a more direct proof, [3] recently improved the result by a factor $O(n)$ for Howard's PI.

**Theorem 2** ([3]). *Howard's PI terminates after at most* $(nm + 1) \left\lceil \frac{1}{1-\gamma} \log \left( \frac{n}{1-\gamma} \right) \right\rceil$ *iterations.*

Our first two results, that are consequences of the contraction properties (Lemmas 2 and 3), are stated in the following theorems.

**Theorem 3** (Proof in Section C). *Howard's PI terminates after at most* $n(m - 1) \left\lceil \frac{1}{1-\gamma} \log \left( \frac{1}{1-\gamma} \right) \right\rceil$ *iterations.*

**Theorem 4** (Proof in Section D). *Simplex-PI terminates after at most* $n(m - 1) \left\lceil \frac{n}{1-\gamma} \log \left( \frac{n}{1-\gamma} \right) \right\rceil$ *iterations.*

Our result for Howard's PI is a factor $O(\log n)$ better than the previous best result of [3]. Our result for Simplex-PI is only very slightly better (by a factor 2) than that of [11], and uses a proof that is more direct. Using more refined argument, we managed to also improve the bound for Simplex-PI by a factor $O(\log n)$.

**Theorem 5** (Proof in Section E). *Simplex-PI terminates after at most* $n^2(m - 1) \left( 1 + \frac{2}{1-\gamma} \log \frac{1}{1-\gamma} \right)$ *iterations.*

Compared to Howard's PI, our bound for Simplex-PI is a factor $O(n)$ larger. However, since one changes only one action per iteration, each iteration may have a complexity lower by a factor $n$: the update of the value can be done in time $O(n^2)$ through the Sherman-Morrisson formula, though in general each iteration of Howard's PI, which amounts to compute the value of some policy that may be arbitrarily different from the previous policy, may require $O(n^3)$ time. Overall, both algorithms seem to have a similar complexity.

It is easy to see that the linear dependency of the bound for Howard's PI with respect to $n$ is optimal. We conjecture that the linear dependency of both bounds with respect to $m$ is also optimal. The dependency with respect to the term $\frac{1}{1-\gamma}$ may be improved, but removing it is impossible for Howard's PI and very unlikely for Simplex-PI. [2] describes an MDP for which Howard's PI requires an exponential (in $n$) number of iterations for $\gamma = 1$ and [5] argued that this holds also when $\gamma$ is in the vicinity of 1. Though a similar result does not seem to exist for Simplex-PI in the literature, [7] consider four variations of PI that all switch one action per iteration, and show through specifically designed MDPs that they may require an exponential (in $n$) number of iterations when $\gamma = 1$.

## 3 Bounds for Simplex-PI that are independent of $\gamma$

In this section, we will describe some bounds that do not depend on $\gamma$ but that will be based on some structural assumptions of the MDPs. On this topic, [8] recently showed the following result for deterministic MDPs.

**Theorem 6** ([8]). *If the MDP is deterministic, then Simplex-PI terminates after at most* $O(n^5 m^2 \log^2 n)$ *iterations.*

Given a policy $\pi$ of a deterministic MDP, states are either on cycles or on paths induced by $\pi$. The core of the proof relies on the following lemmas that altogether show that cycles are created regularly and that significant progress is made every time a new cycle appears; in other words, significant progress is made regularly.

**Lemma 4.** *If the MDP is deterministic, after at most* $nm \lceil 2(n-1) \log n \rceil$ *iterations, either Simplex-PI finishes or a new cycle appears.*

**Lemma 5.** *If the MDP is deterministic, when Simplex-PI moves from $\pi$ to $\pi'$ where $\pi'$ involves a new cycle, we have*

$$\mathbb{1}^T (v_{\pi_*} - v_{\pi'}) \leq \left( 1 - \frac{1}{n} \right) \mathbb{1}^T (v_{\pi_*} - v_\pi).$$

Indeed, these observations suffice to prove[7] that Simplex-PI terminates after $O(n^4 m^2 \log \frac{n}{1-\gamma}) = \tilde{O}(n^4 m^2)$. Removing completely the dependency with respect to the discount factor $\gamma$—the term in $O(\log \frac{1}{1-\gamma})$—requires a careful extra work described in [8], which incurs an extra term of order $O(n \log(n))$.

At a more technical level, the proof of [8] critically relies on some properties of the vector $x_\pi = (I - \gamma P_\pi^T)^{-1} \mathbb{1}$ that provides a discounted measure of state visitations along the trajectories induced by a policy $\pi$ starting from a uniform distribution:

$$\forall i \in X, \quad x_\pi(i) = n \sum_{t=0}^{\infty} \gamma^t \mathbb{P}(i_t = i \mid i_0 \sim U, \ a_t = \pi(i_t)),$$

where $U$ denotes the uniform distribution on the state space $X$. For any policy $\pi$ and state $i$, we trivially have $x_\pi(i) \in \left(1, \frac{n}{1-\gamma}\right)$. The proof exploits the fact that $x_\pi(i)$ belongs to the set $(1, n)$ when $i$ is on a path of $\pi$, while $x_\pi(i)$ belongs to the set $(\frac{1}{1-\gamma}, \frac{n}{1-\gamma})$ when $i$ is on a cycle of $\pi$. As we are going to show, it is possible to extend the proof of [8] to stochastic MDPs. Given a policy $\pi$ of a stochastic MDP, states are either in *recurrent classes* or *transient classes* (these two categories respectively generalize those of cycles and paths). We will consider the following structural assumption.

**Assumption 1.** *Let $\tau_t \geq 1$ and $\tau_r \geq 1$ be the smallest constants such that for all policies $\pi$ and all states $i$,*

$$(1 \ \leq \ )x_\pi(i) \ \leq \ \tau_t \qquad\qquad \textit{if } i \textit{ is transient for } \pi, \textit{ and}$$

$$\frac{n}{(1-\gamma)\tau_r} \ \leq \ x_\pi(i) \left( \ \leq \ \frac{n}{1-\gamma} \right) \qquad\qquad \textit{if } i \textit{ is recurrent for } \pi.$$

The constant $\tau_t$ (resp. $\tau_r$) can be seen as a measure of the time needed to leave transient states (resp. the time needed to revisit states in recurrent classes). In particular, when $\gamma$ tends to 1, it can be seen that $\tau_t$ is an upper bound of the expected time $\mathcal{L}$ needed to "Leave the set of transient states", since for any policy $\pi$,

$$\lim_{\gamma \to 1} \tau_t \geq \frac{1}{n} \lim_{\gamma \to 1} \sum_{i \text{ transient for } \pi} x_\pi(i) = \sum_{t=0}^{\infty} \mathbb{P}(i_t \text{ transient for } \pi \mid i_0 \sim U, \ a_t = \pi(i_t))$$

$$= \mathbb{E}\left[ \ \mathcal{L} \mid i_0 \sim U, \ a_t = \pi(i_t) \right].$$

Similarly, when $\gamma$ is in the vicinity of 1, $\frac{1}{\tau_r}$ is the minimal asymptotic frequency[8] in recurrent states given that one starts from a random uniform state, since for any policy $\pi$ and recurrent state $i$:

$$\lim_{\gamma \to 1} \frac{1-\gamma}{n} x_\pi(i) = \lim_{\gamma \to 1} (1-\gamma) \sum_{t=0}^{\infty} \gamma^t \mathbb{P}(i_t = i \mid i_0 \sim U, \ a_t = \pi(i_t))$$

$$= \lim_{T \to \infty} \frac{1}{T} \sum_{t=0}^{T-1} \mathbb{P}(i_t = i \mid i_0 \sim U, \ a_t = \pi(i_t)).$$

With Assumption 1 in hand, we can generalize Lemmas 4-5 as follows.

**Lemma 6.** *If the MDP satisfies Assumption 1, after at most $n\left[(m-1)\lceil n\tau_t \log(n\tau_t)\rceil + \lceil n\tau_t \log(n^2\tau_t)\rceil\right]$ iterations either Simplex-PI finishes or a new recurrent class appears.*

$$\frac{1}{\tau_r} = \min_{\pi, \ i \text{ recurrent for } \pi} \nu_\pi(i).$$

**Lemma 7.** *If the MDP satisfies Assumption 1, when Simplex-PI moves from $\pi$ to $\pi'$ where $\pi'$ involves a new recurrent class, we have*

$$\mathbb{1}^T(v_{\pi_*} - v_{\pi'}) \leq \left(1 - \frac{1}{\tau_r}\right) \mathbb{1}^T(v_{\pi_*} - v_\pi).$$

From these generalized observations, we can deduce the following original result.

**Theorem 7** (Proof in Appendix F of the Supp. Material)**.** *If the MDP satisfies Assumption 1, then Simplex-PI terminates after at most*

$$n^2(m-1)\left(\lceil \tau_r \log(n\tau_r) \rceil + \lceil \tau_r \log(n\tau_t) \rceil\right)\left[(m-1)\lceil n\tau_t \log(n\tau_t) \rceil + \lceil n\tau_t \log(n^2\tau_t) \rceil\right]$$

*iterations.*

**Remark 1.** *This new result is a strict generalization of the result for deterministic MDPs. Indeed, in the deterministic case, we have $\tau_t \leq n$ and $\tau_r \leq n$, and it is is easy to see that Lemmas 6, 7 and Theorem 7 respectively imply Lemmas 4, 5 and Theorem 6.*

An immediate consequence of the above result is that Simplex-PI is *strongly polynomial* for sets of MDPs that are much larger than the deterministic MDPs mentionned in Theorem 6.

**Corollary 2.** *For any family of MDPs indexed by $n$ and $m$ such that $\tau_t$ and $\tau_r$ are polynomial functions of $n$ and $m$, Simplex-PI terminates after a number of steps that is polynomial in $n$ and $m$.*

# 4    Similar results for Howard's PI?

One may then wonder whether similar results can be derived for Howard's PI. Unfortunately, and as quickly mentionned by [8], the line of analysis developped for Simplex-PI does not seem to adapt easily to Howard's PI, because simultaneously switching several actions can interfere in a way that the policy improvement turns out to be small. We can be more precise on what actually breaks in the approach we have described so far. On the one hand, it is possible to write counterparts of Lemmas 4 and 6 for Howard's PI (see Appendix G of the Supp. Material).

**Lemma 8.** *If the MDP is deterministic, after at most $n$ iterations, either Howard's PI finishes or a new cycle appears.*

**Lemma 9.** *If the MDP satisfies Assumption 1, after at most $nm\lceil \tau_t \log n\tau_t \rceil$ iterations, either Howard's PI finishes or a new recurrent class appears.*

However, on the other hand, we did not manage to adapt Lemma 5 nor Lemma 7. In fact, it is unlikely that a result similar to that of Lemma 5 will be shown to hold for Howard's PI. In a recent deterministic example due to [4] to show that Howard's PI may require at most $O(n^2)$ iterations, new cycles are created every single iteration but the sequence of values satisfies[9] for all iterations $k < \frac{n^2}{4} + \frac{n}{4}$ and states $i$,

$$v_*(i) - v_{\pi_{k+1}}(i) \geq \left[1 - \left(\frac{2}{n}\right)^k\right](v_*(i) - v_{\pi_k}(i)).$$

Contrary to Lemma 5, as $k$ grows, the amount of contraction gets (exponentially) smaller and smaller. With respect to Simplex-PI, this suggests that Howard's PI may suffer from subtle specific pathologies. In fact, the problem of determining the number of iterations required by Howard's PI has been challenging for almost 30 years. It was originally identified as an open problem by [10]. In the simplest—deterministic—case, the question is still open: the currently best known lower bound is the $O(n^2)$ bound by [4] we have just mentionned, while the best known upper bound is $O(\frac{m^n}{n})$ (valid for all MDPs) due to [6].

On the positive side, an adaptation of the line of proof we have considered so far can be carried out under the following assumption.

**Assumption 2.** *The state space $X$ can be partitioned in two sets $\mathcal{T}$ and $\mathcal{R}$ such that for all policies $\pi$, the states of $\mathcal{T}$ are transient and those of $\mathcal{R}$ are recurrent.*

Indeed, under this assumption, we can prove for Howard's PI a variation of Lemma 7 introduced for Simplex-PI.

**Lemma 10.** *For an MDP satisfying Assumptions 1-2, suppose Howard's PI moves from $\pi$ to $\pi'$ and that $\pi'$ involves a new recurrent class. Then*

$$\mathbb{1}^T(v_{\pi_*} - v_{\pi'}) \leq \left(1 - \frac{1}{\tau_r}\right)\mathbb{1}^T(v_{\pi_*} - v_\pi).$$

And we can deduce the following original bound (that also applies to Simplex-PI).

**Theorem 8** (Proof in Appendix H of the Supp. Material). *If the MDP satisfies Assumptions 1-2, then Howard's PI terminates after at most $n(m-1)\left(\lceil \tau_t \log n\tau_t \rceil + \lceil \tau_r \log n\tau_r \rceil\right)$ iterations, while Simplex-PI terminates after at most $n(m-1)\left(\lceil n\tau_t \log n\tau_t \rceil + \lceil \tau_r \log n\tau_r \rceil\right)$ iterations.*

It should however be noted that Assumption 2 is rather restrictive. It implies that the algorithms converge on the recurrent states independently of the transient states, and thus the analysis can be decomposed in two phases: 1) the convergence on recurrent states and then 2) the convergence on transient states (given that recurrent states do not change anymore). The analysis of the first phase (convergence on recurrent states) is greatly facilitated by the fact that in this case, a new recurrent class appears every single iteration (this is in contrast with Lemmas 4, 6, 8 and 9 that were designed to show under which conditions cycles and recurrent classes are created). Furthermore, the analysis of the second phase (convergence on transient states) is similar to that of the discounted case of Theorems 3 and 4. In other words, if this last result sheds some light on the practical efficiency of Howard's PI and Simplex-PI, a general analysis of Howard's PI is still largely open, and constitutes our main future work.

# A   Contraction property for Howard's PI (Proof of Lemma 2)

For any $k$, using the facts that $\{\forall \pi, \; T_\pi v_\pi = v_\pi\}$, $\{T_{\pi_*} v_{\pi_{k-1}} \leq T_{\pi_k} v_{\pi_{k-1}}\}$ and $\{$Lemma 1 and $P_{\pi_k}$ is positive definite$\}$, we have

$$v_{\pi_*} - v_{\pi_k} = T_{\pi_*} v_{\pi_*} - T_{\pi_*} v_{\pi_{k-1}} + T_{\pi_*} v_{\pi_{k-1}} - T_{\pi_k} v_{\pi_{k-1}} + T_{\pi_k} v_{\pi_{k-1}} - T_{\pi_k} v_{\pi_k}$$
$$\leq \gamma P_{\pi_*}(v_{\pi_*} - v_{\pi_{k-1}}) + \gamma P_{\pi_k}(v_{\pi_{k-1}} - v_{\pi_k}) \leq \gamma P_{\pi_*}(v_{\pi_*} - v_{\pi_{k-1}}).$$

Since $v_{\pi_*} - v_{\pi_k}$ is non negative, we can take the max norm and get: $\|v_{\pi_*} - v_{\pi_k}\|_\infty \leq \gamma \|v_{\pi_*} - v_{\pi_{k-1}}\|_\infty$.

# B   Contraction property for Simplex-PI (Proof of Lemma 3)

By using the fact that $\{v_\pi = T_\pi v_\pi \Rightarrow v_\pi = (I - \gamma P_\pi)^{-1} r_\pi\}$, we have that for all pairs of policies $\pi$ and $\pi'$.

$$v_{\pi'} - v_\pi = (I - \gamma P_{\pi'})^{-1} r_{\pi'} - v_\pi = (I - \gamma P_{\pi'})^{-1}(r_{\pi'} + \gamma P_{\pi'} v_\pi - v_\pi)$$
$$= (I - \gamma P_{\pi'})^{-1}(T_{\pi'} v_\pi - v_\pi). \tag{1}$$

On the one hand, by using this lemma and the fact that $\{T_{\pi_{k+1}} v_{\pi_k} - v_{\pi_k} \geq 0\}$, we have for any $k$: $v_{\pi_{k+1}} - v_{\pi_k} = (I - \gamma P_{k+1})^{-1}(T_{\pi_{k+1}} v_{\pi_k} - v_{\pi_k}) \geq T_{\pi_{k+1}} v_{\pi_k} - v_{\pi_k}$, which implies that

$$\mathbb{1}^T(v_{\pi_{k+1}} - v_{\pi_k}) \geq \mathbb{1}^T(T_{\pi_{k+1}} v_{\pi_k} - v_{\pi_k}). \tag{2}$$

On the other hand, using Equation (1) and the facts that $\{\|(I - \gamma P_{\pi_*})^{-1}\|_\infty = \frac{1}{1-\gamma}$ and $(I - \gamma P_{\pi_*})^{-1}$ is positive definite$\}$, $\{\max_s T_{\pi_{k+1}} v_{\pi_k}(s) = \max_{s,\tilde{\pi}} T_{\tilde{\pi}} v_{\pi_k}(s)\}$ and

$\{\forall x \geq 0, \ \max_s x(s) \leq \mathbb{1}^T x\}$, we have:

$$v_{\pi_*} - v_{\pi_k} = (I - \gamma P_{\pi_*})^{-1}(T_{\pi_*} v_{\pi_k} - v_{\pi_k}) \leq \frac{1}{1-\gamma} \max_s T_{\pi_*} v_{\pi_k}(s) - v_{\pi_k}(s)$$

$$\leq \frac{1}{1-\gamma} \max_s T_{\pi_{k+1}} v_{\pi_k}(s) - v_{\pi_k}(s) \leq \frac{1}{1-\gamma} \mathbb{1}^T (T_{\pi_{k+1}} v_{\pi_k} - v_{\pi_k}),$$

which implies (using $\{\forall x, \ \mathbb{1}^T x \leq n\|x\|_\infty\}$) that

$$\mathbb{1}^T (T_{\pi_{k+1}} v_{\pi_k} - v_{\pi_k}) \geq (1-\gamma)\|v_{\pi_*} - v_{\pi_k}\|_\infty \geq \frac{1-\gamma}{n} \mathbb{1}^T (v_{\pi_*} - v_{\pi_k}). \qquad (3)$$

Combining Equations (2) and (3), we get:

$$\mathbb{1}^T (v_{\pi_*} - v_{\pi_{k+1}}) = \mathbb{1}^T (v_{\pi_*} - v_{\pi_k}) - \mathbb{1}^T (v_{\pi_{k+1}} - v_{\pi_k})$$

$$\leq \mathbb{1}^T (v_{\pi_*} - v_{\pi_k}) - \frac{1-\gamma}{n} \mathbb{1}^T (v_{\pi_*} - v_{\pi_k}) = \left(1 - \frac{1-\gamma}{n}\right) \mathbb{1}^T (v_{\pi_*} - v_{\pi_k}).$$

## C  A bound for Howard's PI when $\gamma < 1$ (Proof of Theorem 3)

For any $k$, by using Equation (1) and the fact $\{v_* - v_{\pi_k} \geq 0$ and $P_{\pi_k}$ positive definite$\}$, we have:

$$v_* - T_{\pi_k} v_* = (I - \gamma P_{\pi_k})(v_* - v_{\pi_k}) \leq v_* - v_{\pi_k}.$$

Since $v_* - T_{\pi_k} v_*$ is non negative, we can take the max norm and, using Lemma 2, Equation (1) and the fact that $\{\|(I - \gamma P_{\pi_0})^{-1}\|_\infty = \frac{1}{1-\gamma}\}$, we get:

$$\|v_* - T_{\pi_k} v_*\|_\infty \leq \|v_* - v_{\pi_k}\|_\infty \leq \gamma^k \|v_{\pi_*} - v_{\pi_0}\|_\infty$$

$$= \gamma^k \|(I - \gamma P_{\pi_0})^{-1}(v_* - T_{\pi_0} v_*)\|_\infty \leq \frac{\gamma^k}{1-\gamma} \|v_* - T_{\pi_0} v_*\|_\infty. \qquad (4)$$

By definition of the max-norm, there exists a state $s_0$ such that $v_*(s_0) - [T_{\pi_0} v_*](s_0) = \|v_* - T_{\pi_0} v_*\|_\infty$. From Equation (4), we deduce that for all $k$,

$$v_*(s_0) - [T_{\pi_k} v_*](s_0) \leq \|v_* - T_{\pi_k} v_*\|_\infty \leq \frac{\gamma^k}{1-\gamma} \|v_* - T_{\pi_0} v_*\|_\infty = \frac{\gamma^k}{1-\gamma}(v_*(s_0) - [T_{\pi_0} v_*](s_0)).$$

As a consequence, the action $\pi_k(s_0)$ must be different from $\pi_0(s_0)$ when $\frac{\gamma^k}{1-\gamma} < 1$, that is for all values of $k$ satisfying $k \geq k^* = \left\lceil \frac{\log \frac{1}{1-\gamma}}{1-\gamma} \right\rceil > \left\lceil \frac{\log \frac{1}{1-\gamma}}{\log \frac{1}{\gamma}} \right\rceil$. In other words, if some policy $\pi$ is not optimal, then one of its non-optimal actions will be eliminated for good after at most $k^*$ iterations. By repeating this argument, one can eliminate all non-optimal actions (they are at most $n(m-1)$), and the result follows.

## D  A bound for Simplex-PI when $\gamma < 1$ (Proof of Theorem 4)

Using $\{\forall x \geq 0, \|x\|_\infty \leq \mathbb{1}^T x\}$, Lemma 3, $\{\forall x, \ \mathbb{1}^T x \leq n\|x\|_\infty\}$, Equation (1) and $\{\|(I - \gamma P_{\pi_0})^{-1}\|_\infty = \frac{1}{1-\gamma}\}$, we have for all $k$,

$$\|v_{\pi_*} - T_{\pi_k} v_{\pi_*}\|_\infty \leq \|v_{\pi_*} - v_{\pi_k}\|_\infty \leq \mathbb{1}^T (v_{\pi_*} - v_{\pi_k})$$

$$\leq \left(1 - \frac{1-\gamma}{n}\right)^k \mathbb{1}^T (v_{\pi_*} - v_{\pi_0}) \leq n \left(1 - \frac{1-\gamma}{n}\right)^k \|v_{\pi_*} - v_{\pi_0}\|_\infty$$

$$= n \left(1 - \frac{1-\gamma}{n}\right)^k \|(I - \gamma P_{\pi_0})^{-1}(v_* - T_{\pi_0} v_*)\|_\infty \leq \frac{n}{1-\gamma} \left(1 - \frac{1-\gamma}{n}\right)^k \|v_{\pi_*} - T_{\pi_0} v_{\pi_*}\|_\infty$$

Similarly to the proof for Howard's PI, we deduce that a non-optimal action is eliminated after at most $k^* = \left\lceil \frac{n}{1-\gamma} \log \frac{n}{1-\gamma} \right\rceil \geq \left\lceil \frac{\log \frac{n}{1-\gamma}}{\log\left(1 - \frac{1-\gamma}{n}\right)} \right\rceil$, and the overall number of iterations is obtained by noting that there are at most $n(m-1)$ non optimal actions to eliminate.

## Footnotes

[1] In the works of [11, 8, 3] that we reference, the integer "$m$" denotes the total number of actions, that is $nm$ with our notation. When we restate their result, we do it with our own notation, that is we replace their $"m"$ by $"nm"$.

[2]Restricting our attention to stationary deterministic policies is not a limitation. Indeed, for the optimality criterion to be defined soon, it can be shown that there exists at least one stationary deterministic policy that is optimal [9].

[3]In this case, PI is equivalent to running the simplex algorithm with the highest-pivot rule on a linear program version of the MDP problem [11].

[4]For clarity, all proofs are deferred to the Appendix. The first proofs about bounds for the case $\gamma < 1$ are given in the Appendix of the paper. The other proofs, that are more involved, are provided in the Supplementary Material.

[5]A sequence of non-negative numbers $(x_k)_{k \geq 0}$ is contracting with coefficient $\alpha$ if and only if for all $k \geq 0$, $x_{k+1} \leq \alpha x_k$.

[6]For Howard's PI, we have: $\|v_* - v_{\pi_k}\|_\infty \leq \gamma^k \|v_* - v_{\pi_0}\|_\infty \leq \gamma^k V_{\max}$. Thus, a sufficient condition for $\|v_* - v_{\pi_k}\|_\infty < \epsilon$ is $\gamma^k V_{\max} < \epsilon$, which is implied by $k \geq \frac{\log \frac{V_{\max}}{\epsilon}}{1-\gamma} > \frac{\log \frac{V_{\max}}{\epsilon}}{\log \frac{1}{\gamma}}$. For Simplex-PI, we have $\|v_* - v_{\pi_k}\|_\infty \leq \|v_* - v_{\pi_k}\|_1 \leq \left(1 - \frac{1-\gamma}{n}\right)^k \|v_* - v_{\pi_0}\|_1 \leq \left(1 - \frac{1-\gamma}{n}\right)^k n V_{\max}$, and the conclusion is similar to that for Howard's PI.

[7] This can be done by using arguments similar to the proof of Theorem 4 in Section D.

[8] If the MDP is aperiodic and irreducible, and thus admits a stationary distribution $\nu_\pi$ for any policy $\pi$, one can see that

[9]This MDP has an even number of states $n = 2p$. The goal is to minimize the long term expected cost. The optimal value function satisfies $v_*(i) = -p^N$ for all $i$, with $N = p^2 + p$. The policies generated by Howard's PI have values $v_{\pi_k}(i) \in (p^{N-k-1}, p^{N-k})$. We deduce that for all iterations $k$ and states $i$, $\frac{v_*(i)-v_{\pi_{k+1}}(i)}{v_*(i)-v_{\pi_k}(i)} \geq \frac{1+p^{-k-2}}{1+p^{-k}} = 1 - \frac{p^{-k}-p^{-k-2}}{1+p^{-k}} \geq 1 - p^{-k}(1-p^{-2}) \geq 1 - p^{-k}$.

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
