[Supplementary Material]



# E Another bound for Simplex-PI when $\gamma < 1$ (Proof of Theorem 5)

This second bound for Simplex-PI is a factor $O(\log n)$ better, but requires a slightly more careful analysis.

At each iteration $k$, let $s_k$ be the state in which an action is switched. We have (by definition of the algorithm):

$$T_{\pi_{k+1}} v_{\pi_k}(s_k) - v_{\pi_k}(s_k) = \max_{\pi, s} T_\pi v_{\pi_k}(s) - v_{\pi_k}(s).$$

Starting with arguments similar to those for the contraction property of Simplex-PI, we have:

$$v_{\pi_*} - v_{\pi_k} = (I - \gamma P_{\pi_*})^{-1}(T_{\pi_*} v_{\pi_k} - v_{\pi_k}) \qquad\qquad \text{\{Lemma 1\}}$$

$$\leq \frac{1}{1-\gamma} \max_s T_{\pi_*} v_{\pi_k}(s) - v_{\pi_k}(s) \qquad \{\|(I - \gamma P_{\pi_*})^{-1}\|_\infty = \frac{1}{1-\gamma} \text{ and } (I - \gamma P_{\pi_*})^{-1} \succ 0\}$$

$$\leq \frac{1}{1-\gamma}(T_{\pi_{k+1}} v_{\pi_k}(s_k) - v_{\pi_k}(s_k)), \qquad\qquad \text{\{By definition of } s_k\}$$

which implies that

$$\|v_{\pi_*} - v_{\pi_k}\|_\infty \leq \frac{1}{1-\gamma}(T_{\pi_{k+1}} v_{\pi_k}(s_k) - v_{\pi_k}(s_k)). \tag{5}$$

On the other hand, we have:

$$v_{\pi_{k+1}} - v_{\pi_k} = (I - \gamma P_{\pi_{k+1}})^{-1}(T_{\pi_{k+1}} v_{\pi_k} - v_{\pi_k}) \qquad\qquad \text{\{Lemma 1\}}$$

$$\geq T_{\pi_{k+1}} v_{\pi_k} - v_{\pi_k}, \qquad\qquad \{(I - \gamma P_{\pi_{k+1}})^{-1} \succ 0 \text{ and } T_{\pi_{k+1}} v_{\pi_k} - v_{\pi_k} \geq 0\}$$

which implies that

$$v_{\pi_{k+1}}(s_k) - v_{\pi_k}(s_k) \geq T_{\pi_{k+1}} v_{\pi_k}(s_k) - v_{\pi_k}(s_k). \tag{6}$$

Write $\Delta_k = v_{\pi_*} - v_{\pi_k}$. From Equations (5) and (6), we deduce that:

$$\Delta_{k+1}(s_k) \leq \Delta_k(s_k) - (1-\gamma)\|\Delta_k\|_\infty$$

$$= \left(1 - (1-\gamma)\frac{\|\Delta_k\|_\infty}{\Delta_k(s_k)}\right)\Delta_k(s_k).$$

This implies in particular that

$$\Delta_{k+1}(s_k) \leq \gamma \Delta_k(s_k),$$

but also—since $\Delta_k(s_k)$ and $\Delta_{k+1}(s_k)$ are non-negative—that

$$\|\Delta_k\|_\infty \leq \frac{1}{1-\gamma}\Delta_k(s_k).$$

Now, write $n_k$ the vector on the state space such that $n_k(s)$ is the number of times state $s$ has been switched until iteration $k$ (including $k$). Since by Lemma 1 the sequence $(\Delta_k)_{k\geq 0}$ is non-increasing, we have

$$\|\Delta_k\|_\infty \leq \frac{1}{1-\gamma}\Delta_k(s_k)$$

$$\leq \frac{\gamma^{n_{k-1}(s_k)}}{1-\gamma}\Delta_0(s_k)$$

$$\leq \frac{\gamma^{n_{k-1}(s_k)}}{1-\gamma}\|\Delta_0\|_\infty.$$

At any iteration $k$, let $s_k^* = \arg\max_s n_{k-1}(s)$. Since at each iteration $k$, one of the $n$ component is increased by 1, we necessarily have

$$n_{k-1}(s_k^*) \geq \left\lfloor \frac{k-1}{n} \right\rfloor.$$

Write $k^* \leq k-1$ the last iteration when the state $s_k^*$ was updated, such that we have

$$n_{k-1}(s_k^*) = n_{k^*-1}(s_{k^*}).$$

Since $(\|\Delta_k\|_\infty)_{k \geq 0}$ is non-increasing (using again Lemma 1), we have

$$\|\Delta_k\|_\infty \leq \|\Delta_{k^*}\|_\infty$$
$$\leq \frac{\gamma^{n_{k^*-1}(s_{k^*})}}{1-\gamma}\|\Delta_0\|_\infty$$
$$= \frac{\gamma^{n_{k-1}(s_k^*)}}{1-\gamma}\|\Delta_0\|_\infty$$
$$\leq \frac{\gamma^{\left\lfloor \frac{k-1}{n} \right\rfloor}}{1-\gamma}\|\Delta_0\|_\infty.$$

We are now ready to finish the proof. We have

$$\|v_{\pi_*} - T_{\pi_k}v_{\pi_*}\|_\infty \leq \|\Delta_k\|_\infty$$
$$\leq \frac{\gamma^{\left\lfloor \frac{k-1}{n} \right\rfloor}}{1-\gamma}\|\Delta_0\|_\infty$$
$$\leq \frac{\gamma^{\left\lfloor \frac{k-1}{n} \right\rfloor}}{(1-\gamma)^2}\|v_{\pi_*} - T_{\pi_0}v_{\pi_*}\|_\infty.$$

Using the relation $n\left\lfloor \frac{k-1}{n} \right\rfloor \geq k-n$ and arguments similar to the previous proofs, we deduce that a non-optimal action is eliminated after at most $n\left(1 + \frac{2}{1-\gamma}\log\frac{1}{1-\gamma}\right)$ iterations, and the result follows from the fact that there are at most $n(m-1)$ non-optimal actions.

# F   A general bound for Simplex-PI (Proof of Theorem 7)

The proof we give here is strongly inspired by that for the deterministic case of [8]: the steps (a series of lemmas) are similar. There are essentially two differences. First our arguments are somewhat more direct in that we do not refer to linear programming. Second, it is more general: for any policy $\pi$, we need to consider the set of transient states (respectively recurrent classes) instead of the set of path states (respectively cycles).

For any policy $\pi$, write $\mathcal{R}(\pi)$ for the set of states that are recurrent for $\pi$. Recall that $x_\pi = (I - \gamma P_\pi{}^T)^{-1}\mathbb{1}$. A useful corollary of Lemma 1 is that for any pair of policies $\pi$ and $\pi'$,

$$\mathbb{1}^T(v_{\pi'} - v_\pi) = x_{\pi'}{}^T(T_{\pi'}v_\pi - v_\pi). \tag{7}$$

With some slight abuse of notation, we will write that $s \in \mathcal{R}(\pi)$ if there exists a recurrent class $R \in \mathcal{R}(\pi)$ that contains $s$. We will repeatedly exploit Assumption 1, that we restate here for clarity:

$$\forall s \in \mathcal{R}(\pi), \frac{n}{(1-\gamma)\tau_r} \leq x_\pi(s) \leq \frac{n}{1-\gamma}, \tag{8}$$

$$\forall s \notin \mathcal{R}(\pi), x_\pi(s) \leq \tau_t. \tag{9}$$

As mentioned, before, the proof is structured in two steps: first, we will show that recurrent classes are created often; then we will show that significant progress is made every time a new recurrent class appears.

## F.1   Part 1: Recurrent classes are created often

**Lemma 11.** *Suppose one moves from policy $\pi$ to policy $\pi'$ without creating any recurrent class. Let $\pi_\dagger$ be the final policy before either a new recurrent class appears or the algorithm terminates. Then*

$$\mathbb{1}^T(v_{\pi_\dagger} - v_{\pi'}) \leq \left(1 - \frac{1}{n\tau_t}\right)\mathbb{1}^T(v_{\pi_\dagger} - v_\pi).$$

*Proof.* The arguments are similar to those for the proof of Theorem 4. On the one hand, we have:

$$\mathbb{1}^T(v_{\pi'} - v_\pi) \geq \mathbb{1}^T(T_{\pi'}v_\pi - v_\pi). \tag{10}$$

On the other hand, we have

$$\mathbb{1}^T(v_{\pi_\dagger} - v_\pi) = \mathbb{1}^T(I - \gamma P_{\pi_\dagger})^{-1}(T_{\pi_\dagger}v_\pi - v_\pi)$$

$$= x_{\pi_\dagger}^T(T_{\pi_\dagger}v_\pi - v_\pi)$$

$$= \sum_{s \notin \mathcal{R}(\pi_\dagger)} x_{\pi_\dagger}(s)(T_{\pi_\dagger}v_\pi(s) - v_\pi(s)) + \sum_{s \in \mathcal{R}(\pi_\dagger)} x_{\pi_\dagger}(s)(T_{\pi_\dagger}v_\pi(s) - v_\pi(s))$$

$$\leq n\tau_t \max_{s \notin \mathcal{R}(\pi_\dagger)} T_{\pi_\dagger}v_\pi(s) - v_\pi(s) + \frac{n^2}{1-\gamma} \max_{s \in \mathcal{R}(\pi_\dagger)} T_{\pi_\dagger}v_\pi(s) - v_\pi(s). \qquad \{\text{Equations (8)-(9)}\}$$

Since by assumption cycles of $\pi_\dagger$ are also cycles of $\pi$, we deduce that for all $s \in \mathcal{R}(\pi_\dagger)$, $\pi_\dagger(s) = \pi(s)$, so that $\max_{s \in \mathcal{R}(\pi_\dagger)} T_{\pi_\dagger}v_\pi(s) - v_\pi(s) = 0$. Thus, the second term of the above r.h.s. is null and

$$\mathbb{1}^T(v_{\pi_\dagger} - v_\pi) \leq n\tau_t \max_s T_{\pi_\dagger}v_\pi(s) - v_\pi(s)$$

$$\leq n\tau_t \max_s T_{\pi'}v_\pi(s) - v_\pi(s) \qquad \{\max_s T_{\pi'}v_\pi(s) = \max_{s,\tilde{\pi}} T_{\tilde{\pi}}v_\pi(s)\}$$

$$= n\tau_t \mathbb{1}^T(T_{\pi'}v_\pi - v_\pi). \tag{11}$$

Combining Equations (10) and (11), we get:

$$\mathbb{1}^T(v_{\pi_\dagger} - v_{\pi'}) = \mathbb{1}^T(v_{\pi_\dagger} - v_\pi) - \mathbb{1}^T(v_{\pi'} - v_\pi)$$

$$\leq \left(1 - \frac{1}{n\tau_t}\right)\mathbb{1}^T(v_{\pi_\dagger} - v_\pi).$$

$\square$

**Lemma 12.** *While Simplex-PI does not create any recurrent class nor finishes:*

- *an action is eliminated from policies after at most $\lceil n\tau_t \log(n\tau_t) \rceil$ iterations;*
- *a recurrent class is broken after at most $\lceil n\tau_t \log(n^2\tau_t) \rceil$ iterations.*

*Proof.* Let $\pi$ be the policy in some iteration, $\pi_\dagger$ be the last policy before a new recurrent class appears, and $\pi'$ any policy between $\pi$ and $\pi_\dagger$. Since

$$0 \leq \mathbb{1}^T(v_{\pi_\dagger} - v_\pi) \qquad \{v_{\pi_\dagger} \geq v_\pi\}$$

$$= x_\pi^T(v_{\pi_\dagger} - T_\pi v_{\pi_\dagger}) \qquad \{\text{Equation (7)}\}$$

$$= \sum_{s \notin \mathcal{R}(\pi)} x_\pi(s)(v_{\pi_\dagger}(s) - T_\pi v_{\pi_\dagger}(s)) + \sum_{C \in \mathcal{R}(\pi)} \sum_{s \in C} x_\pi(s)(v_{\pi_\dagger}(s) - T_\pi v_{\pi_\dagger}(s)),$$

there must exist either a state $s_0 \notin \mathcal{R}(\pi)$ such that

$$x_\pi(s_0)(v_{\pi_\dagger}(s_0) - T_\pi v_{\pi_\dagger}(s_0)) \geq \frac{1}{n} x_\pi^T(v_{\pi_\dagger} - T_\pi v_{\pi_\dagger}) \geq 0. \tag{12}$$

or a recurrent class $R_0$ such that

$$\sum_{s \in R_0} x_\pi(s)(v_{\pi_\dagger}(s) - T_\pi v_{\pi_\dagger}(s)) \geq \frac{1}{n} x_\pi^T(v_{\pi_\dagger} - T_\pi v_{\pi_\dagger}) \geq 0. \tag{13}$$

We consider these two cases separately below.

- **case 1:** Equation (12) holds for some $s_0 \notin \mathcal{R}(\pi)$. If $\pi'(s_0) = \pi(s_0)$, then

$$\mathbb{1}^T(v_{\pi_\dagger} - v_{\pi'}) \geq v_{\pi_\dagger}(s_0) - v_{\pi'}(s_0) \qquad \{v_{\pi_\dagger} \geq v_{\pi'}\}$$

$$= v_{\pi_\dagger}(s_0) - T_{\pi'}v_{\pi'}(s_0) \qquad \{v_{\pi'} = T_{\pi'}v_{\pi'}\}$$

$$\geq v_{\pi_\dagger}(s_0) - T_{\pi'}v_{\pi_\dagger}(s_0) \qquad \{v_{\pi_\dagger} \geq v_{\pi'}\}$$

$$= v_{\pi_\dagger}(s_0) - T_\pi v_{\pi_\dagger}(s_0) \qquad \{\pi(s_0) = \pi'(s_0)\}$$

$$\geq \frac{1}{\tau_t} x_\pi(s_0)(v_{\pi_\dagger}(s_0) - T_\pi v_{\pi_\dagger}(s_0)) \qquad \{\text{Equation (9)}\}$$

$$\geq \frac{1}{n\tau_t} x_\pi^T(v_{\pi_\dagger} - T_\pi v_{\pi_\dagger}) \qquad \{\text{Equation (12)}\}$$

$$= \frac{1}{n\tau_t} \mathbb{1}^T(v_{\pi_\dagger} - v_\pi). \qquad \{\text{Equation (7)}\}$$

If there is no recurrent class creation, the contraction property given in Lemma 11 implies that after $k = \lceil n\tau_t \log(n\tau_t) \rceil > \frac{\log(n\tau_t)}{\log \frac{1}{1 - \frac{1}{n\tau_t}}}$ iterations we have

$$\mathbb{1}^T(v_{\pi_\dagger} - v_{\pi'}) < \frac{1}{n\tau_t} \mathbb{1}^T(v_{\pi_\dagger} - v_\pi),$$

and thus $\pi'(s_0) \neq \pi(s_0)$.

- **case 2:** Equation (13) holds for some $R_0 \in \mathcal{R}(\pi)$. Write $\mathcal{T}$ be the set of states that are transient for $\pi$ (formally, $\mathcal{T} = X \backslash \mathcal{R}(\pi)$). For any subset $Y$ of the state space $X$, write $P_\pi^Y$ for the stochastic matrix of which the $i^{th}$ row is equal to that of $P_\pi$ if $i \in Y$, and is 0 otherwise, and write $\mathbb{1}_Y$ the vectors of which the $i^{th}$ component is equal to 1 if $i \in Y$ and 0 otherwise. Using the fact that $P_\pi^{R_0} P_\pi^{\mathcal{T}} = 0$, one can first observe that

$$(I - \gamma P_\pi^{R_0})(I - \gamma P_\pi^{\mathcal{T}}) = I - \gamma(P_\pi^{R_0} + P_\pi^{\mathcal{T}}),$$

from which we can deduce that

$$\begin{aligned}
\mathbb{1}_{\mathcal{T} \cup R_0}{}^T(I - \gamma P)^{-1} &= \mathbb{1}_{\mathcal{T} \cup R_0}{}^T(I - \gamma(P_\pi^{R_0} + P_\pi^{\mathcal{T}}))^{-1} \\
&= \mathbb{1}_{\mathcal{T} \cup R_0}{}^T(I - \gamma P_\pi^{\mathcal{T}})^{-1}(I - \gamma P_\pi^{R_0})^{-1}.
\end{aligned} \tag{14}$$

Also, writing $h_\mathcal{T} = (I - \gamma P_\pi^{\mathcal{T}}{}^T)^{-1}\mathbb{1}_\mathcal{T}$, that satisfies

$$h_\mathcal{T} = \mathbb{1}_\mathcal{T} + \gamma P_\pi^{\mathcal{T}}{}^T h_\mathcal{T},$$

we can see that:

$$\forall s \in R_0, \ h_\mathcal{T}(s) = \gamma \sum_{s' \in \mathcal{T}} p_{s's}(\pi(s')) h_\mathcal{T}(s'), \qquad \{s \in R_0 \ \Rightarrow \ \mathbb{1}_\mathcal{T}(s) = 0\} \tag{15}$$

and thus:

$$(I - \gamma P_\pi^{\mathcal{T}}{}^T)^{-1}\mathbb{1}_{\mathcal{T} \cup R_0}(s) = (I - \gamma P_\pi^{\mathcal{T}}{}^T)^{-1}\mathbb{1}_\mathcal{T}(s) + 1 \qquad\qquad \{P_\pi^{\mathcal{T}}{}^T \mathbb{1}_{R_0} = 0\}$$

$$\begin{aligned}
&= h_\mathcal{T}(s) + 1 \\
&\leq \gamma \sum_{s' \in \mathcal{T}} p_{s's}(\pi(s')) h_\mathcal{T}(s') + 1 \qquad &\{\text{Equation (15)}\} \\
&\leq \sum_{s' \in \mathcal{T}} h_\mathcal{T}(s') + 1 \\
&= \sum_{s' \in \mathcal{T}} x_\pi(s') + 1 \qquad &\{\forall s' \in \mathcal{T}, \ h_\mathcal{T}(s') = x_\pi(s')\} \\
&\leq (n-1)\tau_t + 1 \qquad &\{|\mathcal{T}| \leq (n-1) \text{ and Equation (9)}\} \\
&\leq n\tau_t. \qquad &\{\tau_t \geq 1\} \\
&&(16)
\end{aligned}$$

Writing $\delta$ the vector that equals $v_{\pi_\dagger} - T_\pi v_{\pi_\dagger}$ on $R_0$ and that is null everywhere else, we have

$$\sum_{s \in R_0} x_\pi(s)(v_{\pi_\dagger}(s) - T_\pi v_{\pi_\dagger}(s))$$

$$= \sum_{s \in R_0} [(I - \gamma P_\pi^T)^{-1}\mathbb{1}](s)\delta(s)$$

$$= \sum_{s \in R_0} [(I - \gamma P_\pi^T)^{-1}\mathbb{1}_{\mathcal{T} \cup R_0}](s)\delta(s) \qquad\qquad \left\{\forall s \in R_0,\ [(I - \gamma P_\pi^T)^{-1}\mathbb{1}_{X \setminus (\mathcal{T} \cup R_0)}(s) = 0\right\}$$

$$= \sum_s [(I - \gamma P_\pi^T)^{-1}\mathbb{1}_{\mathcal{T} \cup R_0}](s)\delta(s) \qquad\qquad \{\forall s \notin R_0,\ \delta(s) = 0\}$$

$$= \mathbb{1}_{\mathcal{T} \cup R_0}{}^T (I - \gamma P_\pi)^{-1}\delta$$

$$= \mathbb{1}_{\mathcal{T} \cup R_0}{}^T (I - \gamma P_\pi^{\mathcal{T}})^{-1}(I - \gamma P_\pi^{R_0})^{-1}\delta \qquad\qquad \{\text{Equation (14)}\}$$

$$= \sum_s [(I - \gamma P_\pi^{\mathcal{T}}{}^T)^{-1}\mathbb{1}_{\mathcal{T} \cup R_0}](s)[(I - \gamma P_\pi^{R_0})^{-1}\delta](s)$$

$$= \sum_{s \in R_0} [(I - \gamma P_\pi^{\mathcal{T}}{}^T)^{-1}\mathbb{1}_{\mathcal{T} \cup R_0}](s)[(I - \gamma P_\pi^{R_0})^{-1}\delta](s) \qquad\qquad \{\forall s \notin R_0,\ \delta(s) = 0\}$$

$$= \sum_{s \in R_0} [(I - \gamma P_\pi^{\mathcal{T}}{}^T)^{-1}\mathbb{1}_{\mathcal{T} \cup R_0}](s)(v_{\pi_\dagger}(s) - v_\pi(s)) \qquad\qquad \{\text{Lemma 1}\}$$

$$\leq n\tau_t \mathbb{1}_{R_0}(v_{\pi_\dagger} - v_\pi). \qquad\qquad \begin{matrix}\{\text{Equation (16)}\}\\(17)\end{matrix}$$

Now, one can deduce from this that if $R_0$ is also a recurrent class of $\pi'$, which implies $\mathbb{1}_{R_0}{}^T v_\pi = \mathbb{1}_{R_0}{}^T v_{\pi'}$, then

$$\mathbb{1}^T(v_{\pi_\dagger} - v_{\pi'}) \geq \mathbb{1}_{R_0}{}^T(v_{\pi_\dagger} - v_{\pi'}) \qquad\qquad \{v_{\pi_\dagger} \geq v_{\pi'}\}$$

$$= \mathbb{1}_{R_0}{}^T(v_{\pi_\dagger} - v_\pi) \qquad\qquad \{\mathbb{1}_{R_0}{}^T v_\pi = \mathbb{1}_{R_0}{}^T v_{\pi'}\}$$

$$\geq \frac{1}{n\tau_t}\sum_{s \in R_0} x_\pi(s)(v_{\pi_\dagger}(s) - T_\pi v_{\pi_\dagger}(s)) \qquad\qquad \{\text{Equation (17)}\}$$

$$\geq \frac{1}{n^2\tau_t}x_\pi{}^T(v_{\pi_\dagger} - T_\pi v_{\pi_\dagger}) \qquad\qquad \{\text{Equation (13)}\}$$

$$= \frac{1}{n^2\tau_t}\mathbb{1}^T(v_{\pi_\dagger} - v_\pi). \qquad\qquad \{\text{Equation (7)}\}$$

If there is no recurrent class creation, the contraction property given in Lemma 11 implies that after $k = \lceil n\tau_t \log(n^2\tau_t)\rceil > \frac{\log(n^2\tau_t)}{\log\frac{1}{1-\frac{1}{n\tau_t}}}$ iterations we have

$$\mathbb{1}^T(v_{\pi_\dagger} - v_{\pi'}) < \frac{1}{n^2\tau_t}\mathbb{1}^T(v_{\pi_\dagger} - v_\pi),$$

and thus $R_0$ cannot be a recurrent class of $\pi'$.

$\square$

A direct consequence of the above result is Lemma 6 that we originally stated page 5, and that we restate for clarity.

**Lemma 6.** *After at most $n\left[(m-1)\lceil n\tau_t \log(n\tau_t)\rceil + \lceil n\tau_t \log(n^2\tau_t)\rceil\right]$ iterations, either Simplex-PI finishes or a new recurrent class appears.*

*Proof.* Before a recurrent class is created, at most $n$ recurrent classes need to be broken and $n(m-1)$ actions to be eliminated, and the time required by these events is bounded thanks to the previous lemma. $\square$

## F.2 Part 2: A new recurrent class implies a significant step towards the optimal value

We now proceed to the second part of the proof, and begin by proving Lemma 7 (originally stated page 6).

**Lemma 7.** *Suppose Simplex-PI moves from $\pi$ to $\pi'$ and that $\pi'$ involves a new recurrent class. Then*

$$\mathbb{1}^T(v_{\pi_*} - v_{\pi'}) \leq \left(1 - \frac{1}{\tau_r}\right)\mathbb{1}^T(v_{\pi_*} - v_\pi).$$

*Proof.* Let $s_0$ be the state such that $\pi'(s) \neq \pi(s)$. On the one hand, since $\pi'$ contains a new recurrent class $R$ (necessarily containing $s_0$), we have

$$
\begin{aligned}
\mathbb{1}^T(v_{\pi'} - v_\pi) &= {x_{\pi'}}^T(T_{\pi'}v_\pi - v_\pi) &&\{\text{Equation (7)}\} \\
&= x_{\pi'}(s_0)(T_{\pi'}v_\pi(s_0) - v_\pi(s_0)) &&\{\text{Simplex-PI switches 1 action}\} \\
&\geq \frac{n}{(1-\gamma)\tau_r}(T_{\pi'}v_\pi(s_0) - v_\pi(s_0)). &&\{\text{Equation 8 with } s_0 \in R \subset \mathcal{R}(\pi')\} \quad (18)
\end{aligned}
$$

On the other hand,

$$
\begin{aligned}
v_{\pi_*} - v_\pi &= (I - \gamma P_{\pi_*})^{-1}(T_{\pi_*}v_\pi - v_\pi) &&\{\text{Lemma 1}\} \\
&\leq \frac{1}{1-\gamma}\max_s T_{\pi_*}v_\pi(s) - v_\pi(s) &&\{\|(I - \gamma P_{\pi_*})^{-1}\|_\infty \leq \frac{1}{1-\gamma} \text{ and } (I - \gamma P_{\pi_*})^{-1} \succ 0\} \\
&\leq \frac{1}{1-\gamma}\max_s T_{\pi'}v_\pi(s) - v_\pi(s) &&\{\max_s T_{\pi'}v_\pi(s) = \max_{s,\bar\pi} T_{\bar\pi}v_\pi(s)\} \\
&= \frac{1}{1-\gamma}(T_{\pi'}v_\pi(s_0) - v_\pi(s_0)). &&\{\text{Simplex-PI switches 1 action}\}\quad(19)
\end{aligned}
$$

Combining these two observations, we obtain:

$$
\begin{aligned}
\mathbb{1}^T(v_{\pi_*} - v_{\pi'}) &= \mathbb{1}^T(v_{\pi_*} - v_{\pi'}) - \mathbb{1}^T(v_{\pi'} - v_\pi) \\
&\leq \mathbb{1}^T(v_{\pi_*} - v_{\pi'}) - \frac{n}{(1-\gamma)\tau_r}(T_{\pi'}v_\pi(s_0) - v_\pi(s_0)) &&\{\text{Equation (18)}\} \\
&\leq \mathbb{1}^T(v_{\pi_*} - v_{\pi'}) - \frac{n}{\tau_r}\max_s v_{\pi*}(s) - v_{\pi'}(s) &&\{\text{Equation (19)}\} \\
&\leq \left(1 - \frac{1}{\tau_r}\right)\mathbb{1}^T(v_{\pi_*} - v_{\pi'}). &&\{\forall x, \ \mathbb{1}^T x \leq n\max_s x(s)\}
\end{aligned}
$$

$\square$

**Lemma 14.** *While the algorithm does not terminate,*

- *some non-optimal action is eliminated from recurrent states after at most $\lceil \tau_r \log(n\tau_r)\rceil$ recurrent class creations;*

- *some non-optimal action is eliminated from policies after at most $\lceil \tau_r \log(n\tau_t)\rceil$ recurrent class creations.*

*Proof.* Let $\pi$ be the policy in some iteration and $\pi'$ be any policy between $\pi$ and $\pi_*$. Let $s_0 = \arg\max_s x_\pi(s)(v_{\pi_*}(s) - T_\pi v_{\pi_*}(s))$. We have

$$
\begin{aligned}
x_\pi(s_0)(v_{\pi_*}(s_0) - T_\pi v_{\pi_*}(s_0)) &\geq \frac{1}{n}{x_\pi}^T(v_{\pi_*} - T_\pi v_{\pi_*}) &&\{\forall x, \ \mathbb{1}^T x \leq n\max_s x(s)\} \\
&= \mathbb{1}^T(v_{\pi_*} - v_\pi). &&\{\text{Equation (7)}\} \quad (20)
\end{aligned}
$$

We now consider two cases.

- **case 1:** $s_0 \notin \mathcal{R}(\pi)$. If $\pi'(s_0) = \pi(s_0)$, then

$$
\begin{aligned}
\mathbb{1}^T(v_{\pi_*} - v_{\pi'}) &= {x_{\pi'}}^T(v_{\pi_*} - T_{\pi'}v_{\pi_*}) &&\{\text{Equation (7)}\} \\
&\geq x_{\pi'}(s_0)(v_{\pi_*}(s_0) - T_{\pi'}v_{\pi_*}(s_0)) &&\{v_{\pi_*} \geq T_{\pi'}v_{\pi_*}\} \\
&\geq v_{\pi_*}(s_0) - T_{\pi'}v_{\pi_*}(s_0) &&\{x_{\pi'}(s_0) \geq 1\} \\
&= v_{\pi_*}(s_0) - T_\pi v_{\pi_*}(s_0) &&\{\pi(s_0) = \pi'(s_0)\} \\
&\geq \frac{1}{\tau_t}x_\pi(s_0)(v_{\pi_*}(s_0) - T_\pi v_{\pi_*}(s_0)) &&\{\text{Equation (9)}\} \\
&\geq \frac{1}{n\tau_t}\mathbb{1}^T(v_{\pi_*} - v_\pi). &&\{\text{Equation (20)}\}
\end{aligned}
$$

After $k = \lceil \tau_r \log n\tau_t \rceil > \frac{\log n\tau_t}{\log \frac{1}{1-\frac{1}{\tau}r}}$ new recurrent classes are created, we have by the contraction property of Lemma 7 that

$$\mathbb{1}^T(v_{\pi_*} - v_{\pi'}) < \frac{1}{n\tau_t}\mathbb{1}^T(v_{\pi_*} - v_\pi).$$

This implies that $\pi'(s_0) \neq \pi(s_0)$.

- **case 2:** $s_0 \in \mathcal{R}(\pi)$. If $\pi'(s_0) = \pi(s_0)$ and $s_0 \in \mathcal{R}(\pi')$, then

$$
\begin{aligned}
\mathbb{1}^T(v_{\pi_*} - v_{\pi'}) &= x_{\pi'}{}^T(v_{\pi_*} - T_{\pi'}v_{\pi_*}) && \{\text{Equation (7)}\}\\
&= \sum_s x_{\pi'}(s)(v_{\pi_*}(s) - T_{\pi'}v_{\pi_*}(s))\\
&\geq \sum_{s \in R_0} x_{\pi'}(s)(v_{\pi_*}(s) - T_{\pi'}v_{\pi_*}(s)) && \{v_{\pi_*} \geq T_{\pi'}v_{\pi_*}\}\\
&\geq \frac{n}{(1-\gamma)\tau_r}\sum_{s \in R_0} v_{\pi_*}(s) - T_{\pi'}v_{\pi_*}(s) && \{\text{Equation 8}\}\\
&\geq \frac{n}{(1-\gamma)\tau_r}v_{\pi_*}(s_0) - T_{\pi'}v_{\pi_*}(s_0) && \{v_{\pi_*} \geq T_{\pi'}v_{\pi_*}\}\\
&= \frac{n}{(1-\gamma)\tau_r}v_{\pi_*}(s_0) - T_{\pi}v_{\pi_*}(s_0) && \{\pi(s_0) = \pi'(s_0)\}\\
&= \frac{1}{\tau_r}x_\pi(s_0)(v_{\pi_*}(s_0) - T_\pi v_{\pi_*}(s_0)) && \{x_\pi(s_0) \leq \frac{n}{1-\gamma}\}\\
&\geq \frac{1}{n\tau_r}\mathbb{1}^T(v_{\pi_*} - v_\pi). && \{\text{Equation (20)}\}
\end{aligned}
$$

After $k = \lceil \tau_r \log n\tau_r \rceil > \frac{\log n\tau_r}{\log \frac{1}{1-\frac{1}{\tau}r}}$ new recurrent classes are created, we have by the contraction property of Lemma 7 that

$$\mathbb{1}^T(v_{\pi_*} - v_{\pi'}) < \frac{1}{n\tau_r}\mathbb{1}^T(v_{\pi_*} - v_\pi).$$

This implies that $\pi'(s_0) \neq \pi(s_0)$ if $s_0$ is recurrent for $\pi'$.

$\square$

We are ready to conclude: At most, the $n(m-1)$ non-optimal actions may need to be eliminated from recurrent and transient states, requiring at most a total of $n(m-1)(\lceil \tau_r \log(n\tau_r)\rceil + \lceil \tau_r \log(n\tau_t)\rceil)$ recurrent classes creations. The result follows from the fact that each class creation requires at most $n\left[(m-1)\lceil n\tau_t \log(n\tau_t)\rceil + \lceil n\tau_t \log(n^2\tau_t)\rceil\right]$ iterations.

# G  Cycle and recurrent classes creations for Howard's PI (Proofs of Lemmas 8 and 9)

**Lemma 8.** *If the MDP is deterministic, after at most $n$ iterations, either Howard's PI finishes or a new cycle appears.*

*Proof.* Consider a sequence of $l$ generated policies $\pi_1, \cdots, \pi_l$ from an initial policy $\pi_0$ such that no new cycle appears. By induction, we have

$$
\begin{aligned}
v_{\pi_l} - v_{\pi_k} &= T_{\pi_l}v_{\pi_l} - T_{\pi_l}v_{\pi_{k-1}} + T_{\pi_l}v_{\pi_{k-1}} - T_{\pi_k}v_{\pi_{k-1}} + T_{\pi_k}v_{\pi_{k-1}} - T_{\pi_k}v_{\pi_k} && \{\forall \pi,\ T_\pi v_\pi = v_\pi\}\\
&\leq \gamma P_{\pi_l}(v_{\pi_l} - v_{\pi_{k-1}}) + \gamma P_{\pi_k}(v_{\pi_{k-1}} - v_{\pi_k}) && \{T_{\pi_l}v_{\pi_{k-1}} \leq T_{\pi_k}v_{\pi_{k-1}}\}\\
&\leq \gamma P_{\pi_l}(v_{\pi_l} - v_{\pi_{k-1}}). && \{\text{Lemma 1 and } P_{\pi_k} \succ 0\}\\
&\leq (\gamma P_{\pi_l})^k(v_{\pi_l} - v_{\pi_0}). && \{\text{by induction on } k\}
\end{aligned}
$$

$$(21)$$

Since the MDP is deterministic and has $n$ states, $(P_{\pi_l})^n$ will only have non-zero values on columns that correspond to $\mathcal{R}(\pi_l)$. Furthermore, since no cycle is created, $\mathcal{R}(\pi_l) \subset \mathcal{R}(\pi_0)$, which implies that $v_{\pi_l}(s) - v_{\pi_0}(s) = 0$ for all $s \in \mathcal{R}(\pi_l)$. As a consequence, we have $(P_{\pi_l})^n (v_{\pi_l} - v_{\pi_0}) = 0$. By Equation (21), this implies that $v_{\pi_l} = v_{\pi_n}$. If $l > n$, then the algorithm must have terminated. $\quad\square$

**Lemma 9.** *If the MDP satisfies Assumption 1, after at most $nm\lceil \tau_t \log n\tau_t \rceil$ iterations, either Howard's PI finishes or a new recurrent class appears.*

*Proof.* It can be seen that the proof of Lemma 6 also applies to Howard's PI. $\quad\square$

# H  A bound for Howard's PI and Simplex-PI  under Assumption 2 (proof of Theorem 8)

We here consider that the state space is decomposed into 2 sets: $\mathcal{T}$ is the set of states that are transient under all policies, and $\mathcal{R}$ is the set of states that are recurrent under all policies. From this assumption, it can be seen that when running Howard's PI or Simplex-PI, the values and actions chosen on $\mathcal{T}$ have no influence on the evolution of the values and policies on $\mathcal{R}$. So we will study the convergence of both algorithms in two steps: We will first bound the number of iterations to converge on $\mathcal{R}$. We will then add the number of iterations for converging on $\mathcal{T}$ given that convergence has occurred on $\mathcal{R}$.

**Convergence on the set $\mathcal{R}$ of recurrent states**  Without loss of generality, we consider that the state space is only made of the set of recurrent states.

First consider Simplex-PI. If all states are recurrent, new recurrent classes are created at every iteration, and Lemma 6 holds. Then, in a way similar to the proof of Lemma 14, it can be shown that every $\lceil \tau_r \log n\tau_r \rceil$ iterations, a non-optimal action can be eliminated. As there are at most $n(m-1)$ non-optimal actions, we deduce that Simplex-PI converges in at most $n(m-1)\lceil \tau_r \log n\tau_r \rceil$ iterations on $\mathcal{R}$.

Now consider Howard's PI. We can prove Lemma 10, that we restate for clarity.

**Lemma 10.** *For an MDP satisfying Assumptions 1-2, suppose Howard's PI moves from $\pi$ to $\pi'$ and that $\pi'$ involves a new recurrent class. Then*

$$\mathbb{1}^T (v_{\pi_*} - v_{\pi'}) \leq \left(1 - \frac{1}{\tau_r}\right) \mathbb{1}^T (v_{\pi_*} - v_\pi).$$

*Proof.* In the case we focus on the convergence on the set $\mathcal{R}$ of recurrent states, new recurrent classes are created at every iteration. So we will prove that the inequality holds for every $k$. On the one hand, we have for all iterations $k$,

$$\mathbb{1}^T (v_{\pi_{k+1}} - v_{\pi_k}) = x_{\pi_{k+1}}{}^T (T_{\pi_{k+1}} v_{\pi_k} - v_{\pi_k}) \qquad \{\text{Equation (7)}\}$$

$$\geq \frac{n}{(1-\gamma)\tau_r} \mathbb{1}^T (T_{\pi_{k+1}} v_{\pi_k} - v_{\pi_k} \qquad \{\text{Equation (8)}\}$$

$$\geq \frac{n}{(1-\gamma)\tau_r} \|T_{\pi_{k+1}} v_{\pi_k} - v_{\pi_k}\|_\infty. \qquad \{\forall x \geq 0, \mathbb{1}^T x \geq \|x\|_\infty\} \qquad (22)$$

On the other hand,

$$\mathbb{1}^T (v_{\pi_*} - v_{\pi_k}) = x_{\pi_*}{}^T (T_{\pi_*} v_{\pi_k} - v_{\pi_k}) \qquad \{\text{Equation (7)}\}$$

$$\leq \frac{n}{1-\gamma} \|T_{\pi_*} v_{\pi_k} - v_{\pi_k})\|_\infty \qquad \{\sum_i x_{\pi_*}(i) \leq \frac{n}{1-\gamma}\}$$

$$\leq \frac{n}{1-\gamma} \|T_{\pi_{k+1}} v_{\pi_k} - v_{\pi_k}\|_\infty. \qquad (23)$$

By combining Equations (22) and (23), we obtain:

$$\mathbb{1}^T (v_{\pi_*} - v_{\pi_{k+1}}) = \mathbb{1}^T (v_{\pi_*} - v_{\pi_k}) - \mathbb{1}^T (v_{\pi_{k+1}} - v_{\pi_k})$$

$$\leq \left(1 - \frac{1}{\tau_r}\right) \mathbb{1}^T (v_{\pi_*} - v_{\pi_k}).$$

$$\square$$

Then, similarly to Simplex-PI, we can prove that after every $\lceil \tau_r \log n\tau_r \rceil$ iterations a non-optimal action must be eliminated. And as there are at most $n(m-1)$ non-optimal actions, we deduce that Howard's PI converges in at most $n(m-1)\lceil \tau_r \log n\tau_r \rceil$ iterations on $\mathcal{R}$.

**Convergence on the set $\mathcal{T}$ of transient states** Consider now that convergence has occurred on the recurrent states $\mathcal{R}$. A simple variation of the proof of Lemma 6/Lemma 9 (where we use the fact that we don't need to consider the events where recurrent classes are broken since recurrent classes do not evolve anymore) allows to show that the extra number of iterations to converge on the transient states is $n(m-1)\lceil \tau_t \log n\tau_t \rceil$ for Howard's PI and $n^2(m-1)\lceil \tau_t \log n\tau_t \rceil$ for Simplex-PI, and the result follows.