[Reviews · NeurIPS 2013]

Submitted by Assigned_Reviewer_4

Summary:
The paper considers the complexity of policy iteration (PI) for finite MDPs
with n states and m actions per state in a number of settings.
The following results are derived:
For Howard's PI, a previous bound is improved by removing a log(n) factor.
For Simplex PI, a previous bound is improved by halving its multiplicative constant.
When the MDP is deterministic, a previous bound for Simplex PI
is improved by decreasing the power of
dependence on n by one.
This result is generalized to stochastic MDPs under additional restrictions on the MDP.

Originality and significance:
The complexity of PI is of major interest to the RL community. Even though the finite
setting is not practical, it is important to have a good understanding of this
simple case before moving to more complicated settings. This paper essentially picks
up the previous line of research that started with a paper by Yinyu Ye in 2011 and
continued by Post and Ye, as well as by Hansen, Miltersen and Zwick.
Some of the improvements are marginal and the core ideas of the proofs go back to the
2011 work of Ye. However, the proofs are much more streamlined and are way more
accessible to RL researchers than those of Ye (who prefers to use the language
of linear programming). The extension of strong polynomial results to stochastic MDPs
under structural assumptions are the most important novel parts of the paper.

Clarity:
The paper is relatively well-written, though the authors should check it carefully
as it contains a few typos (lines 285, 292). Also, at times fluency could be improved
(e.g., "However, on the other hand" on line 303 is awkward).


Soundness:
I believe that the results are sound (I did not have time to check all the details).

Minor issues:
- line 367: specify the lemma used
Summary: The mildly improves previous results on the time complexity of PI,
while making the proofs more accessible to RL researchers.
Perhaps more importantly, the paper shows that the class of MDPs
that simplex PI is strongly
polynomial for is much larger than what was known beforehand.

Submitted by Assigned_Reviewer_5

This is a nice paper that tightens previous bounds on policy iteration, a problem on which recent progress has been made after decades of stagnation. The results are nice, though not shockingly impressive in themselves. Importantly, however, the paper provides short proofs of some key results and good, short summaries with real insight in cases where the proofs aren't short. The paper is extremely well written.

Minor comments:

Line 97.5: at least on state -> at least one state

line 106: all iteration k -> all iterations k

line 114.5: convergen often occur -> convergence often occurs

line 203: on path -> on paths

Summary: Modest improvement on previous policy iteration results. Very nicely written paper, a pleasure to read.

Submitted by Assigned_Reviewer_6

This paper provides a theoretical analysis of two variants of policy iteration.
The analysis improves the known worst-case iteration complexity by O(log(n)) in
one case, and by the constant factor 2 in the other. They generalize their
analysis for one variant run on deterministic MDPs, improving its iteration
complexity by O(n).

I cannot recommend this paper be accepted in its current form. The exposition
is extremely rushed, and the improved bounds in themselves are not impressive.
It is possible that the proof techniques provide some scientific value (as the
authors claim), but this is incredibly difficult to distill from the paper.

The proof techniques that improve the first two bounds follow prior work
closely. The proofs in the deterministic MDP setting appear more novel, but
(a) key terms, like transient and recurrent, are not defined and (b) the
authors state that the assumptions required for their proofs are overly strong.
That is, their correctness and applicability cannot be evaluated.

The authors need to better motivate this work and more clearly
describe/highlight their contributions.

Minor comments:
- please spell check
- line 85, \pi' is not used inside the max
- 115, convergen
- 125, observtation
- 134, mentionned
- footnote 6, not immediate put in appendix
- 177, Sherman-Morrisson
- 180, this statement is not immediate, though I believe it
- 182, explaination for conjectures?
- 233, transient and recurrent need to be defined
- 277, appdx
- 292, mentionned and developped

After viewing author feedback and discussions with other reviewers, it is apparent the initial quality score was overly harsh. I suggest drawing more attention to contribution contained in the proofs themselves in the next revision to aid less astute readers like myself.
Summary: I recommend this paper be rejected due as it does not appear to provide significantly better bounds, or any obviously insightful proof techniques.

Submitted by Assigned_Reviewer_7

The paper presents bounds on the complexity of policy iteration for two versions of the algorithm: one which changes the policy at all states where there is an action better than the current policy (which the author call Howard PI), and one which only changes policy at states with a maximal advantage (which corresponds to running Simplex in the linear programming version of the algorithm). The paper re-derives some existing bounds in a simpler way and also improves some existing bounds.

The paper presents an interesting contribution and the proofs seem correct. The novelty is not very high, as the paper builds on existing results, but nonetheless, there are improvements on these results. The writing in the paper is clear.

Small comments:
- line o43: When at state -> At state
- The definition of advantage (line 085) is missing a \prime, otherwise it's trivially - line 097: on state -> one state
- line 112: and that -> that
- line 195: the the
- line 217: Material and
- line 367: "this lemma" - which lemma do you mean?
- In the proof of Lemma 3, line 72, you have a \max_{s,\bar{\pi}}. Here, the order in which you take the max matters, so you should write \max_s \max_{\bar{\pi}} to clarify (I am assuming this is what is done).
- It would be very useful to have some examples of types of MDPs in which Corollary 2 holds.
Summary: The paper presents improvements on the current bounds of policy iteration, making progress on existing bounds. It is a nice contribution, though it does not depart in a major way from existing proof techniques.
Author Feedback

Author rebuttal: Thanks for the reviews.


A few comments about the last review (Reviewer_6), that is the only one to be negative.

<< (a) key terms, like transient and recurrent, are not defined >>
We believe transience and recurrence are standard for Markov chains, that underlie MDPs.

<< (b) the authors state that the assumptions required for their proofs are overly strong. >>
This is not true. The only assumption that we say is restrictive is the one used for the last theorem. All other assumptions and definitions are, to our opinion, pretty natural. In particular, please reread the end of the paper.

<< It is possible that the proof techniques provide some scientific value (as the
authors claim), but this is incredibly difficult to distill from the paper. >>
Note that the proof of the bound for Howard's PI (the version of PI that most RL researchers think of) with gamma<1 (Appendix A+Appendix C) is sufficiently small/simple to be included in any introductory course about MDPs. This is one of the motivations for including it in extenso in paper, while putting it in a much bigger perspective (recent known results, improvements, and remaining important problems).